# Genomic Space of *MGMT* in Human Glioma Revisited: Novel Motifs, Regulatory RNAs, NRF1, 2, and CTCF Involvement in Gene Expression

**DOI:** 10.3390/ijms22052492

**Published:** 2021-03-02

**Authors:** Mohammed A. Ibrahim Al-Obaide, Viswanath Arutla, Manny D. Bacolod, Wei Wang, Ruiwen Zhang, Kalkunte S. Srivenugopal

**Affiliations:** 1Department of Pharmaceutical Sciences, Jerry H. Hodge School of Pharmacy, Texas Tech University Health Sciences Center, Amarillo, TX 79106, USA; m.alobaide1950@gmail.com (M.A.I.A.-O.); Viswanath.arutla@ttuhsc.edu (V.A.); 2Department of Microbiology and Immunology, Weill Cornell Medicine, New York, NY 10065, USA; mdb2005@med.cornell.edu; 3Department of Pharmacological and Pharmaceutical Sciences, College of Pharmacy, University of Houston, Houston, TX 77204, USA; wwang4@central.uh.edu (W.W.); rzhang27@central.uh.edu (R.Z.)

**Keywords:** *MGMT*, DNA repair, alkylating agents, brain tumors, CTCF, regulatory RNAs

## Abstract

Background: The molecular regulation of increased *MGMT* expression in human brain tumors, the associated regulatory elements, and linkages of these to its epigenetic silencing are not understood. Because the heightened expression or non-expression of *MGMT* plays a pivotal role in glioma therapeutics, we applied bioinformatics and experimental tools to identify the regulatory elements in the *MGMT* and neighboring *EBF3* gene loci. Results: Extensive genome database analyses showed that the MGMT genomic space was rich in and harbored many undescribed RNA regulatory sequences and recognition motifs. We extended the *MGMT*’s exon-1 promoter to 2019 bp to include five overlapping alternate promoters. Consensus sequences in the revised promoter for (a) the transcriptional factors CTCF, NRF1/NRF2, GAF, (b) the genetic switch MYC/MAX/MAD, and (c) two well-defined *p53* response elements in *MGMT* intron-1, were identified. A putative protein-coding or non-coding RNA sequence was located in the extended 3′ UTR of the *MGMT* transcript. Eleven non-coding RNA loci coding for miRNAs, antisense RNA, and lncRNAs were identified in the *MGMT-EBF3* region and six of these showed validated potential for curtailing the expression of both *MGMT* and *EBF3* genes. ChIP analysis verified the binding site in *MGMT* promoter for CTCF which regulates the genomic methylation and chromatin looping. CTCF depletion by a pool of specific siRNA and shRNAs led to a significant attenuation of MGMT expression in human GBM cell lines. Computational analysis of the ChIP sequence data in ENCODE showed the presence of NRF1 in the *MGMT* promoter and this occurred only in MGMT-proficient cell lines. Further, an enforced NRF2 expression markedly augmented the *MGMT* mRNA and protein levels in glioma cells. Conclusions: We provide the first evidence for several new regulatory components in the *MGMT* gene locus which predict complex transcriptional and posttranscriptional controls with potential for new therapeutic avenues.

## 1. Introduction

Human O^6^-methylguanine DNA methyltransferase (MGMT), also called O^6^-alkylguanine DNA alkyltransferase (AGT) is a simple DNA repair protein involved in the protection of the normal cellular genome from the mutagenic actions of alkylating agents [1,2]. The single-step error-free stoichiometric reaction involving the transfer of O^6^-alkylguanines bound to DNA restores the normal G-C base pairing and prevents the GC to AT transitions [3]. MGMT is expressed at different extents in normal tissues ranging from high amounts in the liver to negligible levels in the brain and bone marrow, indicating a tissue-specific expression [1,2,3]. However, human cancers including brain tumors express the MGMT protein in abundance [4,5,6,7]; a large fraction of CNS malignancies, nevertheless are *MGMT*-deficient due to gene silencing by promoter methylation [8,9]. The MGMT status in brain cancers is of huge importance for treatment because the antimutagenic function of MGMT interferes with the cytotoxic actions of anticancer alkylating agents [7]. This is because MGMT effectively repairs the O^6^-methylguanine and O^6^-chloroethylguanine lesions induced by methylating agents (temozolomide, TMZ), and chloroethylating agents (BCNU, CCNU), respectively, thereby preventing the generation of mutagenic lesions and interstrand DNA cross-links [7,10]. Consequently, MGMT has emerged as a central determinant of tumor resistance to clinically used alkylating agents. Both the antimutagenic and drug resistance properties conferred by MGMT derive from its ability to remove the alkyl groups from the O6-position in the DNA to cysteine 145 at the active site in a unique suicidal reaction [1,2,5]. Whether expressed or not expressed, *MGMT* remains a focal point in the translational neuro-oncology forming a bridge between resistance and sensitivity to the alkylating drugs. When expressed at elevated, albeit varied levels as in a majority of brain cancers, it is a major target for pseudosubstrate [11,12] or repurposed inhibitors [13,14] to increase drug efficacy. However, when *MGMT* expression is epigenetically inactivated through promoter methylation in a subset of CNS tumors, it is associated with a better response to chemotherapy, greater overall survival, and longer time to progression [15,16,17,18]. Therefore, testing of gene promoter methylation status has emerged as an important clinical procedure. Other reports point to the important role of gene body methylation in the epigenetic regulation of *MGMT* expression. Several reports have demonstrated the positive correlation between *MGMT* expression and methylation at CpG sites within the gene body [19,20,21].

Human *MGMT* is encoded by a single copy gene localized on the telomeric region of chromosome 10 at 10q26. It is relatively a large gene (~304 Kb) containing long introns and 5 short exons, the first of which is non-coding [22]. The gene has a TATA-less CAAT-less promoter containing a CpG island, which was first characterized in 1991 by Harris et al. [23]. The promoter with maximal activity was shown to lie 5′ of the gene from -953 to +202 bp (with the TSS as +1) comprising the minimal promoter (−69 to +19) [23], enhancer (+143 to +202) [24], to which an enhancer-binding protein binds, and several TF binding sites, such as AP1 and SP1 [25]. The MGMT promoter contains a 777-bp long CpG island with 97 CpG sites [15,18]. *MGMT* promoter methylation is a prototype example of the terms CpG island methylator phenotype (CIMP) that describes the extensive DNA hypermethylation of promoter-associated CpG islands in human cancers [26] and G-CIMP that denotes the CIMP status in human glioma [27]. This epigenetic signature was first described in glioblastoma (GBM) and was later validated in lower-grade gliomas, recurrent GBMs, and pediatric glioma subtypes [27]. The methylation sites and changes in methylation in *MGMT*’s exon-1 region have been extensively studied in cancer cell lines with and without MGMT expression and in primary tumor specimens [18,19,20,21]. In general, the results have been inconsistent indicating that the methylation sites affecting gene expression are not uniformly distributed throughout the CpG islands [19,22]. Studies in cell lines identified two methylated areas in the CpG island: a highly methylated region upstream of the transcription start site (TSS), including a minimal promoter and a highly methylated region downstream of exon 1 [19,20,21,22]. Methylation of CpGs located at the first noncoding exon and enhancer appeared more critical for loss of *MGMT* expression. Therefore, most of the molecular assays in clinical use are designed to interrogate these regions [15,22]. Research in recent years has uncovered the molecular basis of the G-CIMP process. Somatic mutations in isocitrate dehydrogenases (*IDHs*) 1 and 2 and loss-of-function mutations in *ten-elven translocation (TET)-methylcytosine dioxygenase-2 (TET2)* have been shown to confer and maintain the G-CIMP activity [28]. Briefly, the mutant IDH enzymes gain an unusual new activity of reducing the α-ketoglutarate (α-KG) to an oncometabolite, D-2 hydroxyglutarate (D-2HG). D-2HG competes with α-KG and inhibits a large number of α-KG-dependent dioxygenases such as the TET, histone lysine demethylase, and the ALKBH DNA repair proteins that ultimately lead to hypermethylation of the CpG islands in the genome [29]. Consequently, most gliomas positive for *IDH* mutations also exhibit *MGMT* promoter methylation [30]. However, the status of CpG island methylation has remained divisive, as prolonged survival was also reported without promoter hypermethylation [31]. Additionally, the *MGMT* promoter methylation varies widely in gliomas in the range of 35% to 84%. Moreover, promoter methylation does not always show a strong correlation with MGMT protein levels [32].

From the foregoing, it is clear that our knowledge of *MGMT* gene regulation is restricted to the 5′- region involving the promoter and methylation status of the CpG islands therein. The genomic space of *MGMT* is vast and the regulatory elements controlling the transcription and methylation in the other exons and introns are not known. For example, exon 2, where the meaningful transcription start is separated from the non-coding exon 1 by an intronic space of 69 kb, and no data is available on elements associated therewith. With these considerations, we undertook a comprehensive bioinformatic analysis of the *MGMT* and the contiguous *EBF3* genes and report the involvement of CTCF [33,34] and NRF1/NRF2 in the regulation of this unique DNA repair system. The study also uncovered several important motifs including p53 binding sites and regulatory RNAs.

## 2. Results and Discussion

### 2.1. Recapitulation of MGMT Genomic Space and the mRNA

Our bioinformatic studies indicated that the *MGMT*-coding region corresponds to about 0.5% of its genomic space. The MGMT gene is located on the forward (plus) strand of the long arm of chromosome 10 mapped to 10q26.3 at genomic coordinates (GRCh38): chr10:129,467,189–129,768,041. Figure 1A shows the setting of *MGMT* locus with the lengths of exonic and intronic spaces. Note that exon 1 and a large part of exon 5 are non-coding; the sequences of all exons, translated and untranslated, their span, and the corresponding locations in the *MGMT* genome are represented in Appendix A.

The genomic context of the 10q26 region as presented in the GenBank and ensemble databases show a vast genomic region composed of about 500 kb that includes two contiguous genes, *MGMT* and *EBF3* (early B-cell factor), and the intervening space. We designated this area as the *MGMT-EBF3* region and found it to harbor several regulatory sequences and uncharacterized non-coding RNA loci. Since gene expression is often controlled and coregulated at both short stretches and along with large topological domains [35], it was of interest to analyze the genomic and epigenomic regulatory sequences in the *MGMT-EFB3* genomic space (Figure 1B). The *EBF3* resides close to *MGMT* gene, just 67.3 kb away on the reverse (negative) strand and located at chr10:129,835,232–129,964,281 with a length of 129,050 nucleotides. The *EBF* family of genes encode DNA binding transcription factors involved in multiple cellular functions including neurogenesis and bone development promoting a pro-apoptotic function [36]. *EBF3* gene is frequently inactivated in human cancers including glioma, leukemia, and colon cancer [36,37,38,39,40,41,42]. Remarkably, the *EBF* gene is co-methylated with *MGMT* in glioma and other cancers [38,39,41] suggesting a possible regulatory association.

### 2.2. New Features of 3′ Untranslated Region (UTR) of MGMT Transcript

The UTRs in processed gene transcripts play a crucial role in gene expression by influencing the localization, stability, export, and translation efficiency of mRNAs. We used the UTRdb research tool [43] to reexamine the 5’ UTR and 3’ UTR regions of *MGMT* mRNA. The 5’ UTR region was 26 bps long, equivalent to 23% of MGMT Exon-1. Whereas the length of the 3’ UTR region was 522 bps that represents 84% of the exon-5 untranslated region (Figure 2A). Exon-5 is composed of 833 bps that represent 61% of the *MGMT* mRNA transcript, yet the length of the translated sequence of exon-5 is 140 bps, 16.8% of the sequence. The 3’-side of the *MGMT* locus showed an added 3 kb sequence (Figure 2A,B) in the updated version of the GenBank-Gene, which can be considered as an extension of the 3’ UTR region. We noted for the first time that *MGMT*’s extended 3′UTR region is quite novel and bears the signatures of what was thought until recently to be an uncharacterized protein-coding gene but now thought to be a non-coding RNA. Our analysis showed the extra length in the untranslated *MGMT* exon 5 sequence in the recent version of *MGMT* mRNA, accession: NM_002412.5, to be the sequence of an unknown protein-coding locus Loc105378559. This locus is mapped at chr10:129,768,365–129,770,983, with the intergenic space between *MGMT* and the above sequence of just 150 bps. The GenBank-Gene discontinued locus 105378559 on 9 June 2016. Ensembl-85 dated July 2016 showed 105378559 gene, however, discontinued it in October 2016 in the Ensembl-86. The previous versions of the two databases showed the discontinued uncharacterized locus 105378559 with an mRNA of two exons and a predicted protein size of 244 amino acids (26.84 kDa). The Ensembl-86 also showed in the genomic space of the discontinued locus 105378559, an overlapping sequence of a non-coding RNA gene, ENSG00000227374 (RP11-109A6.3), located at chr10: 129,768,844–129,769,435. Indeed, initial work in our laboratory has shown that a protein corresponding to the locus 105378559 exists in human glioma cells (not shown); the possible co-expression of this protein with MGMT and any other regulatory interrelationship between them merits further investigation. The sequence of the 3′ region of the *MGMT* transcript is shown in Figure 2B.

### 2.3. New Regulatory Motifs in MGMT Promoter; ARE, ERE, and NRF1/NRF2 -Mediated Upregulation of MGMT Expression

Once the minimal promoter for *MGMT* was recognized [23], efforts were made to identify and characterize the cis-regulatory elements in the promoter. The glucocorticoid response element [25,44] and seven SP1 binding sites were soon reported [25]. Six SP1 sites, 3 upstream, 3 downstream of the TSS were characterized [25]. We screened the minimal promoter for additional cis sequences/motifs and found the presence of an antioxidant response element (ARE) and estrogen response element (ERE). The locations of the ARE and ERE along with the known GRE, AP1, and SP1 binding sites in the original 1.15 kb promoter are shown in Figure 2C. The ARE is a cis-acting enhancer sequence that mediates transcriptional activation of genes in cells exposed to oxidative stress and chemopreventive agents [45]. The AREs are activated upon binding of the NRF2 (nuclear factor 2-related factor 2) [46]. The AREs have been identified in the regulatory regions of numerous genes that encode cytoprotective enzymes/proteins and are upregulated in response to oxidants, electrophiles, and natural products [46]. These include the *glutathione-S transferases (GSTs), γ-glutamyl cysteinyl synthetase (γ-GCS), NAD(P)H: quinone oxidoreductase (NQO1),* and others. NRF2 is a basic region-leucine zipper (bZIP)-type transcription factor. The consensus ARE sequence, which is also the Nrf2 binding site is TGACnnnGC. We found two of these (ARE1 and -2) in the *MGMT* promoter—between 617–595 and 584–562 positions. Our earlier reports of MGMT upregulation in human tumor cells by antioxidant phytochemicals and a cysteine prodrug [47,48] are indeed suggestive of the DNA repair gene being a target ARE-NRF2 interaction. To further confirm the involvement of NRF2 in *MGMT* expression, we stably transfected the human glioblastoma SF-188 cells with an NRF2 expression vector and determined the *MGMT* mRNA and protein levels (Figure 3A). A significant increase of *MGMT* transcripts and protein (~4-fold) was observed in NRF2 expressing cells compared to the untransfected parent cells (Figure 3A right panel). These observations confirm that MGMT expression can be upregulated during redox stress as a cytoprotective measure through the ARE-NRF2 axis. Regarding the ERE, we previously showed that the motif present in MGMT bound to the estrogen receptor-α in electrophoretic mobility shift assays and that estrogen moderately upregulated MGMT expression in ER-positive MCF-7 cells [49].

Further, we analyzed the ChIP sequence data stored in the ENCODE database using the transcription factor analysis tool Alibaba2.1 (http://generegulation.com/pub/programs/alibaba2/, accessed on 15 February 2021) to look for NRF transcription factors 1 or 2 localized to the *MGMT* promoter. Interestingly, we found the ERE motif located <300 bases upstream of 5′ end of exon 1 (nucleotides 741–746 highlighted in green in Figure 2C) as a recognition site for NRF1. Indeed, the ChIP Seq Data for NRF1 (Abcam’s mouse monoclonal antibody targeting amino acids 201–286 of human NRF1 was used) indicate that NRF1 binds to the *MGMT* promoter region which covers this particular ERE regulatory site (Figure 3B). NRF1 binding was highly evident in the MGMT-proficient cell lines GM12878 (B-lymphocyte), H1-hESC (embryonic stem cells), and HeLa-S3 (cervical carcinoma), but not in MGMT-deficient K562 (chronic myelogenous leukemia) cells. Furthermore, the cell lines GM12878, H1-hESC, and HeLa-S3 exhibited an *MGMT* promoter region with relaxed chromatin (as reflected by a high degree of DNAseI hypersensitivity), and a transcriptionally active state evident by the presence of *MGMT* gene transcripts. In contrast, neither DNaseI hypersensitivity nor MGMT transcription was evident in the cell line K562 (Figure 3B). Overall, these observations suggest NRF1′s likely role in the regulation of MGMT expression.

Both NRF1 and NRF2 are Cap’n’Collar family of transcription factors with overlapping functions in the cellular response to oxidative stress [50]. Both of them can bind the ARE and NRF-1 null mice compensate by activating the expression of NRF-2 dependent gene expression [50,51]. Furthermore, NRF-1 has been shown to bind selectively to the ERE [47] and play a role in estrogen-promoted mitochondrial biogenesis [52]. These observations are consistent with and explain our results showing increased expression of MGMT by NRF2 (Figure 3A) and selective localization of NRF1 on ERE in *MGMT* promoter (Figure 3B).

### 2.4. Revised MGMT Promoter That Includes Alternate Promoters, CGI Status, and Other Promoter-Like Sequences in MGMT Genome

About half of all eukaryotic genes possess alternative promoters, which are thought to add flexibility and diversity to the regulation of gene expression [53]. The functional significance of alternative promoters and their role in physiology and pathology, however, has not been fully defined, except for some genes where these sequences are aberrantly activated, developmentally regulated, or silenced [54]. Our extensive search of databases showed the presence of additional and overlapping promoter sequences on the 5′ side, and all of these were presently extending beyond the currently established minimal promoter (Table 1). The exon-1 and exon-2 hosted a total of six promoters, five overlapping promoters mapped at 5’-side of exon-1 and one predicted promoter-like sequence located at the end of intron-1 adjacent to exon-2 approximately 70 kilobases downstream *MGMT* exon-1. Preliminary analysis of this putative exon-2 promoter showed the abundance of ATF1, TATA, and INR motifs that are features of an active transcription unit. Because exon-2 per se is transcribed into the first coding region for *MGMT*, it is tempting to speculate the second promoter-like sequence may be involved in accelerating the transcription of exons 2 through 5. The database sources, map locations, and the length of each promoter are shown in Table 1A. The combined sequences of the alternative promoters form a sequence composed of 2019 bps mapped at chr10:129466183–129468201 we refer to the revised *MGMT-E1* promoter (P1). We revised the *MGMT-E1* promoter to include the corresponding alternative promoters which consist of 2019 bp. The complete sequence of the revised E-1 promoter is shown in Appendix A. The Eukaryotic Promoter Database (EPD) tool showed 35 transcription start sites (TSSs) positions in the *MGMT* exon-1 alternative promoter, *MGMT*_1, 32 located along upstream region (−1 to −100) compared to two TSS positions at the downstream region, +1 to +7 in addition to TSS at position 0, which was expressed in 593 samples compared to smaller samples in other TSS positions (Appendix A). These observations suggest that alternate promoters may be used in some tissues and may occasionally drive the *MGMT* transcription.

Next, we analyzed the CGI distribution in the revised exon-1 promoter. The CGIs in 2019 bp promoter, designated P1-CGI showed two sequences (upstream and downstream of TSS) characterized by high %G + C and Obs/Exp CpG values. The values observed for upstream CGI (US-CGI) (%G + C = 65.2, Obs/Exp CpG = 1.1), and downstream CGI, DS-CG (%G + C = 73.1, Obs/Exp CpG = 0.79) are shown in Table 2. The DS-CGI of the MGMT-E1 promoter hosts the minimal promoter, TSS site, and CTCF motif, to be discussed later in this report. Interestingly, the two promoter CGI regions, upstream or downstream, are also rich in single nucleotide polymorphisms (SNPs) (not shown). The EMBOSS CPGplot tool was used to compare and quantitate the CGIs. The plot of the CGIs of *MGMT* exon-1 promoter GenBank-nucleotide “X61657.1” composed of 1157 bps hosted CGI of 621 bps (481–1101) (Appendix A). The revised *MGMT-P1* promoter of 2019 bps hosted CGI of 776 bps (476–1251) (Appendix A). While the new sequences do not add additional CGIs, the revised version provides a better prediction for the two sub-CGIs on either side of *MGMT-TSS*.

### 2.5. Recognition Motifs for GAF and MYC/MAD/MAX Switch within the MGMT Promoter

Transcription pausing is a well-known mechanism associated with and promotion of the elongation step in active transcription. The key steps include the positioning of RNA polymerase II (Pol II) in the promoter-proximal region about 30–50 bp near the transcription start site (pausing) and its release for rapid and constructive elongation [55]. The pausing depends on the transcription factor binding motifs of the core promoter and the GAGA associated factor (GAF) which binds to the GAGA sequences in the region. GAF plays a crucial role [55,56,57] in transcription-pausing. We report the presence of three GAF factor motifs (GAGAG and GAGAGA) in the *MGMT-P1* sequence (Figure 4, upper panel). Mutations of the GAGA sequences result in a loss of Pol II pausing [57]. Therefore, we suggest that GAF plays a role in stabilizing a pause in *MGMT* transcription. 

Further, we identified enhancer or E box hexanucleotide, 5′-CACGTG-3′ motifs for the MYC/MAX/MAD network in the *MGMT-P1* promoter (Figure 4 upper panel). The regulatory mechanism of this genetic switch is a result of the network interaction of three basic helix-loop-helix leucine zipper transcription factors, *MYC, MAX,* and *MAD* [58]. Briefly, while the MYC protein is capable of weak homodimerization, optimal MYC function requires heterodimerization with Max. The Myc-Max dimers bind DNA, activate transcription, and promote cell proliferation or apoptosis [58,59]. Max also forms homodimers or heterodimers with its alternative partner, Mad. Max-Mad complexes behave as antagonists of Myc/Max through competition for DNA binding and recruiting repressor complexes containing histone deacetylases [59].

### 2.6. Identification of p53 Response Elements (PREs) in Intron 1

A direct role of tumor suppressor *p53* in the regulation of MGMT expression has been a subject of several studies with consistent evidence pointing to the repression of *MGMT* transcription by wild-type p53 [60,61,62,63]. Thus, our studies using adenoviral *p53* infection of H1299 cells (*p53-null*) [63] and other reports of wild-type *p53* overexpression [56,57] led to a remarkable downregulation of *MGMT* mRNA and protein levels, highly suggesting a repressive role of normal *p53* in controlling MGMT expression. However, the mechanisms of this attenuation and whether the *MGMT* gene possesses p53 binding sites remain unknown. Interestingly, our manual search of the *MGMT* genome for transcription factor sites revealed a consensus p53 binding motif [64] which mediates *p53*-dependent transcriptional responses. In-silico analysis of the *MGMT* intron-1 region composed of 69,056 bps covering the area between *MGMT* exon-1 and exon-2 showed the presence of p53 response elements (PREs). Two PREs were identified, PRE-1 and PRE-2, with a consensus motif composed of two decamer motifs RRRCWWGYYY, where R=A or G; W=A or T; Y=C or T, separated by a spacer of eight and nine bps respectively. Two sequences that harbor PREs were identified in the reverse strand of *MGMT* intron-1. PRE-1 composed of AGGCAAGCCCACACCCAGGCTAGCaC violated PRE-consensus motifs by only one base; the second PRE violated PRE consensus by three bases, AAGCATGCaaAAAGCAAAAACATGTaT, located nearer to the 3’-side of exon-1 (Figure 4, lower panel). While the actual binding of p53 to these sites and the functionality awaits further investigation, it is clear that *MGMT* transcription is under tight control by several regulatory mechanisms involved in the activation, repression, and pausing of *MGMT* transcription.

### 2.7. Presence of CTCF Recognition Motifs in the MGMT Promoter and Rest of the Gene: Evidence for CTCF as a Positive Regulator of MGMT Expression

The CCCTC-binding factor (CTCF) is a conserved 11 zinc finger phosphoprotein and a versatile transcription factor with multiple functions [33,34]. Formation of CTCF– site-specific DNA complexes, many of which are methylation-sensitive, results in distinct functions, including transcriptional activation, repression, imprinting, X chromosome inactivation, and chromatin insulation through blocking of the communication between enhancers and promoters [65]. Another widely recognized function of CTCF is to stabilize and facilitate the formation of short-range and long-range chromatin loops to control the genome-wide organization of chromatin architecture [34,66]. In this process of generating topologically-associating domains (TAD), CTCF complexes, and works with another ring-shaped protein called cohesin, setting up the boundaries between the domains [66,67]. To perform these regulatory functions, CTCF utilizes highly conserved and ubiquitous CTCF DNA binding sites that cover the genome. Because while 15% of CTCF-recognition sites are located near promoters and ~40% are within exons and introns [68], CTCF appears to play multiple roles apart from an enhancer blocking activity. CTCF can bind a histone acetyltransferase (HAT)-containing complex or histone deacetylase (HDAC)-containing assembly and function as a transcriptional activator or repressor respectively [69,70]. Nevertheless, many transcriptional effects exerted by CTCF may be related to its ability to build and maintain chromatin loops in collaboration with cohesin as well.

Noteworthy of discussion here is an intimate association between CTCF and epigenetic regulation and possible implications for *MGMT* promoter methylation. It has been known for many years that CTCF binding is abolished by the DNA methylation of CpG sites within the CTCF motif [69,71]. Cancer genome shows widespread disruption of CTCF binding associated with increased methylation; for example, the gain of DNA methylation is enhanced in the absence of CTCF [69], and CTCF dearth induces progressive gain of CpG Island (CGI) methylation in Rb1 promoter [72]. Abnormal DNA methylation patterns of CTCF-binding sites are also associated with transcriptional regulation of tumor suppressor genes such as the *p16INK4* in human cancers [73]. In this context, our bioinformatic analysis of the human *MGMT* gene showed an unusually high congregation of CTCF-binding sites. Analysis of the *MGMT-E1* promoter sequence using JASPAR and CTCFBSDB 2.0 tools enabled us to identify a CTCF site composed of 19 nucleotides, TTACCTCTAGGTGCCAGCC, at chr10:129467562–129467581, present just downstream of the TSS (Figure 5A). A search of the *MGMT* genome with the Ensembl database showed the presence of 9 other high-affinity CTCF binding sites three in intron 1, 5 in intron 2, and one in exon 5 (Figure 5B). Both tools gave a high score for the predicted CTCF binding motif equal to 12. To validate the CTCF site within the promoter, we performed a ChIP assay in SF-188 GBM cells using primers listed in Materials and Methods. A positive control to detect the presence of RNA polymerase on the promoter of the house-keeping gene *GAPDH* was included in these assays. An expected 80 bp amplicon containing the *CTCF* binding site was observed (Figure 5C) confirming the location of this transcription factor on the *MGMT* promoter for the first time.

Since CTCF functions to modulate transcription, next, we studied the linkage between the MGMT and CTCF protein expression levels in a panel of human brain tumor and other cancer cell lines. Western blotting and densitometric quantitation of protein bands showed a variable expression of these proteins. However, there was no clear or statistically valid correlation between the abundance of the two proteins, with many of the high MGMT expressing cells also showing higher levels of CTCF (Figure 6). The results point to the multifaceted role of CTCF in human cells in gene expression and chromatin organization besides a differential expression of two genes in a given cell type and replication state.

To determine the specific role of CTCF in controlling the *MGMT* transcription, we performed extensive silencing of *CTCF* gene expression in a panel of four glioblastoma GBM6, GBM10, T98G, and SF188 cell lines using a set of three siRNA and four shRNA pools (Appendix A). The targets of siRNA and shRNA were located within five exons of *CTCF* transcript variant 1 (NM_006565.3) which is composed of 12 exons. The map locations of the siRNA and shRNA *CTCF* targeted sequences are shown in Appendix A.

The OriGene’s siRNA-27 kit (Rockville, USA) of three Dicer-Substrate 27-mer duplexes were used in the first approach. The 27-mer sites were selected by a rational design algorithm that ensures that the chosen sites do not target alternatively spliced exons and do not include known SNPs., A scrambled universal negative control RNA duplex that is absent in the human genome was used as a control. The western blots and the repressing influence of *CTCF* siRNA on *MGMT* expression and the densitometric quantitation of the protein bands are shown in Figure 7. The downregulation of CTCF after 48 h of transient transfection of siRNA was consistently accompanied by a decrease in MGMT levels. This was clearly evident in all 4 GBM cell lines with the scrambled siRNA controls showing no significant decrease of MGMT protein. The attenuation of the MGMT protein was roughly proportional to the reduction of CTCF protein. One-way ANOVA statistical analyses verified the differences in the diminished protein levels as significant suggesting a partnership and direct correlation between them.

Transient transfection of a pool of four *CTCF* shRNA plasmids also resulted in a reduction of MGMT protein in the GBM cell lines (Figure 8). The moderate downregulation in contrast to the siRNA experiments (Figure 7) may relate to the 24 h exposure and the need for the generation of shRNA in adequate amounts to silence and deplete the CTCF protein. Nevertheless, the findings confirm that CTCF has a function in maintaining the *MGMT* transcription.

Our findings, however, do not shed light on the exact mechanism(s) by which CTCF participates and promotes *MGMT* transcription. The CTCF binding site at the promoter may have a different function than the motifs present at the introns and exon 5 the of MGMT gene. As discussed earlier, human gliomas positive for IDH mutations manifest a CpG island methylator phenotype (G-CIMP) due to the inhibition of the TET-family of 5′-methylcytosine hydroxylases by the oncometabolite D-2HG [29]. When the IDH mutant gliomas develop increased methylation at CTCF binding sites, CTCF fails to bind to its motifs [69,71]. Given the known fact that cell lines/tumors with MGMT promoter methylation poorly transcribe the gene, it is tempting to suggest that a compromised binding of CTCF envisaged in our RNA interference studies (Figure 7 and Figure 8) may reflect the deficient transcriptional transactions and/or altered enhancer protein binding [74] that occur in cells with MGMT promoter hypermethylation leading to a reduction in MGMT protein levels. As such, subtle changes in CTCF binding can result in activation or inhibition of enhancer activity over a long distance in chromatin. For example, loss of CTCF at a domain boundary has been shown to allow a constitutive enhancer to aberrantly activate the PDGFRA, a prominent glioma oncogene [75]. Based on the known role of CTCF in DNA methylation, it will be important to characterize the contribution of CTCF to the epigenetic silencing of *MGMT* expression in glioma and other cancers. Such endeavors may provide new approaches for *MGMT*-targeted therapeutics.

### 2.8. Potential Involvement of RNA Regulatory Elements (Long Non-Coding RNA, Antisense RNA, Micro RNA) in the MGMT- EBF3 Region in the Regulation of MGMT Expression

The variable expression of *MGMT* mRNA in normal tissues and cancers suggests the presence of posttranslational and tissue-specific regulatory elements within the gene. The involvement of miRNAs (142–3p and 181d and others) in controlling *MGMT* expression has been widely reported [76,77,78,79,80]. In this background, our search using the RNA databases (RNA Central, EMBL-EBI, and others) showed a high prevalence of RNA-based determinants within the *MGMT* and the neighboring *EBF* locus (Figure 5B). Since these elements may regulate the expression of either the *MGMT* or *EBF3* or both genes, we undertook a bioinformatic characterization of the sequences. Non-coding RNAs are a large family of transcripts that are classified according to their length into small (<200 nucleotides) and long (>200 residues) [81]. They are now well established in regulating gene expression, maintaining cellular homeostasis, and functions. These are mRNA-like transcripts, transcribed by RNA polymerase II, but lack stable open reading frames. lncRNAs can regulate gene transcription through Interaction with and recruitment of chromatin-modifying enzymes to the target gene locus, or form RNA–protein complexes (RNPs) and help to change the chromatin architecture. The antisense lncRNAs originate from the complementary strand of protein-coding genes. 

Approximately 50% of miRNAs are produced from non-coding transcripts, and many are embedded in the intronic regions of protein-coding genes [82]. Most miRNAs are transcribed in the form of a primary miRNA (pri-miRNA) by RNA polymerase II (Pol II), then processed by the nuclear microprocessor (comprised by the Ribonuclease II DROSHA, and DGCR8) to form the pre-miRNA, which is later exported to the cytoplasm through an Exportin-5-Ran-GTP-shuttle protein. In the cytoplasm, DICER binds to the pre-miRNA and cleaves it to its final 22 nucleotide mature form that associates with AGO 2 to form the RNA-induced silencing complex (RISC). MiRNAs function through sequence complementary: within the RISC, the miRNA binds the target mRNA 3′UTR and, based on the degree of complementarity, leads to full mRNA degradation or blocking of the ribosomal machinery, both result in gene silencing [83].

Our exhaustive efforts to uncover the genes encoding ncRNA within the *MGMT* locus resulted in the identification of two lncRNA, one antisense RNA, and one miRNA. In addition, seven non-coding RNA genes were identified in the immediate distal EBF3 genomic space and the intervening space between *MGMT* and *EBF3* loci. A general map of all these RNA regulatory elements is represented in Figure 5B. Again, the non-coding RNAs in the *EBF3* and *MGMT-EBF3* genomic space belonged to the miRNA, antisense RNA, and lncRNA categories. Of the multiple ncRNAs found in the *MGMT-EBF2* genomic region, we focused on those with the potential to affect *MGMT* expression. We analyzed the RNA-RNA sequence-specific recognition between ncRNA and *MGMT* mRNA (NM_002412.4) or *EBF3* mRNA (NM_001005463.2) transcripts and looked for homologous pairing between the hairpins formations of the two transcripts (Table 3 and Table 4). These analyses were performed using the RNAhybrid software [84] to predict multiple potential binding sites of miRNAs in the target *MGMT* and *EBF3* transcripts. This program is designed to find the energetically most favorable hybridizations of a small RNA to a large RNA. Intramolecular hybridizations, that is, base pairings between target nucleotides or between miRNA nucleotides are not allowed. The program predicts optimal and additional suboptimal, nonoverlapping hits, up to a user-defined number threshold. Statistical significance of predicted targets (*p*-value) was assessed with extreme value statistics of length normalized minimum free energies, a Poisson approximation of multiple binding sites, and the calculation of effective numbers of orthologous targets in comparative studies of multiple organisms. Additionally, E-value is the “expected number of hits given the size of the database was determined. Using these parameters, we identified the microRNA recognition elements (MREs) of *MGMT* (NM_002412.4) and *EBF3* (NM_001005463.2) mRNA sequences. Table 4 shows the first evidence that four of six analyzed ncRNA transcripts have the potential to interact with the *MGMT* transcript. To highlight and validate the sequence complementarity of the new ncRNAs show with MGMT mRNA, we compared them with two reported microRNAs (MIR127 and MIR370), which are known to downregulate *MGMT* expression [79,85] as shown in Table 4. The overall complementarity scores, *p* and E values calculated for MIR127 and MIR370 were comparable with the newly identified RNA regulatory elements (Table 4). Furthermore, it is noteworthy to point out that MIR127 and MIR370, despite their ability to interfere with *MGMT* transcription, arise elsewhere in the genome and not in the MGMT locus per se, unlike the regulatory sequences reported here. Another highly interesting finding is related to the two miRNA genes, ENSG00000266061 and ENSG00000266676 (miRNA 4297) reported in the Ensembl database; both loci were mapped in the *MGMT-EBF3* region in this report. The two miRNA loci were found in the intragenic regions of *MGMT* and *EBF3* respectively (Figure 5B and Table 4). It is tempting to suggest a potential involvement of the two miRNA loci in *MGMT* expression.

Taken together, we provide the first and strong bioinformatic evidence for the presence of RNA regulatory elements within the *MGMT* and *MGMT-EBF* loci. The comparable statistical values of the new ncRNA sequences vis-à-vis the known miRNAs affecting *MGMT* expression (Table 4) lends support to their functionality. However, experimental evidence proving their involvement in their role in negating DNA repair awaits further studies.

## 3. Conclusions

Much of the *MGMT* transcriptional studies and its negative regulation in gliomas are heavily related to the promoter methylation, however, the regulatory elements elsewhere in the gene have received scant attention. This study provided the first evidence for alternative promoters, new intronic and exonic regulatory motifs, the involvement of the CTCF and NRF2 transcription factors, and several RNA regulatory determinants within the *MGMT* and neighboring genomic region. The presence of GAGA-associated factor binding sites (GAF) and the *MYC/MAX/MAD* genetic switch within the *MGMT-E1* promoter are suggestive of tight control of transcriptional regulation of *MGMT*, reflecting a pause control and a fine interplay of *Myc* gene and its binding partners in *MGMT* expression. While several studies have reported a repressive role for wild-type p53 in *MGMT* expression and higher levels of MGMT in mutant p53 harboring tumor cells [61,63,86], the presence of the p53 consensus motif we report in intron 1 provides a mechanistic platform to investigate *MGMT* governance by the tumor suppressor. Bioinformatic evidence also showed a promoter/enhancer-like sequence proximal of MGMT’s exon 2, which harbors a TATA box but no CpG islands. The significance of this regulatory element for transcription of actual coding sequence (exon 2 through 5) remains to be determined. A significant finding reported here is the involvement of the CTCF transcription factor in promoting *MGMT* transcription. CTCF binding sites were found in the minimum promoter and throughout the *MGMT* gene. We suggest CTCF plays multiple roles in *MGMT* transcription. One of them likely is organizing in the *MGMT* gene region in topologically associated loops/domains (TADs) along with cohesin to mediate short and long-range interactions with the transcription machinery. Since CTCF binding is abolished by DNA methylation of CpG sites within the CTCF motif [69,71] and a CTCF binding site is indeed present in the *MGMT* promoter (Figure 5B,C), it is tempting to speculate that lack of CTCF may promote the methylation- associated repression of MGMT. Again, further work is required to answer this clinically important scenario in gliomas.

*MGMT* is a direct target gene of microRNAs, miR-370-3p, and an enhanced expression of miR-370-3p that occurs in glioblastomas increases the TMZ sensitivity [85]. Additionally, miR-130a was found as a predictive marker for TMZ response in patients with GBM, independently of MGMT status [87]. Other investigators reported that high levels of miR-326/miR-130a and low levels of miR-323/miR-329/miR-155/miR-210 were significantly associated with longer overall survival of GBM patients [88]. In this context, the abundant presence of RNA regulatory elements in the form of micro RNAs, antisense, and LncRNAs within the *MGMT* and the immediately adjacent *EBF2* genes is highly significant and the discovery opens up a new area of investigations relevant to MGMT expression in cancer pathophysiology, drug resistance and treatment of gliomas and other cancer types. The new set of identified regulatory motifs and control elements in MGMT expression promise to provide new insights into transcriptional and posttranscriptional mechanisms associated with the governance of this unique DNA repair gene.

## 4. Materials and Methods

### 4.1. Cell Culture

Three human cell lines used in this study, a pediatric glioma SF188 (Grade 4) purchased from the Department of Neurosurgery, Univ. of California, San Francisco. GBM6 and GBM 10 cells were obtained from Dr. Jann Sarkaria (Mayo Clinic, Rochester, MN). T98G (Grade 4) and SNB19 (highly invasive glioblastoma) and other cancer cell lines were obtained from The American Type Culture Collection (Manassas, VA, USA). Cells were grown in Dulbecco’s modified Eagle’s (DMEM) medium supplemented with 10% Fetal Bovine Serum and antibiotics in 5% CO_2_/humidified atmosphere at 37 °C.

### 4.2. Western Blotting

Cells were trypsinized and washed with PBS before lysis by sonication in the presence of protease inhibitor cocktail and phosphatase inhibitors in 50 mM Tris buffer (pH 8.0) containing 5% glycerol and 0.5 mM EDTA. After protein quantitation by the Bradford method, equivalent protein amounts in different treatments were combined with the SDS sample buffer [50 mM Tris-HCl (pH 6.8), 1% SDS, 40% glycerol, and 0.025% Bromophenol Blue], and heated at 90 °C for 2 min. The samples were electrophoresed on 12% SDS-polyacrylamide gels along with the pre-stained molecular weight markers. After electrophoretic transfer of the proteins onto Immobilon-P membranes (Millipore Co.) and blocking in 4% non-fat dry milk, the blots incubated with anti-MGMT monoclonal antibodies (1 µg/mL) for 8 h followed by secondary antibody linked with horseradish peroxidase. Protein bands were visualized by enhanced chemiluminescence (ECL, Amersham Co. Wauwatosa. WI, USA). The blots were photographed, and the bands quantitated by densitometry using a VersaDoc 5000 imaging system (Bio-rad, Hercules, CA, USA) and ImageJ software.

### 4.3. siRNA and shRNA Transient Transfections and NRF2 Stable Transfection

OriGene’s siRNA-27 kit (Rockville, MD, USA) containing 3 siRNAs targeting a specific CTCF transcript sequence was purchased. The siRNA pool or the scrambled counterparts was transfected at 10 nM using the Roche X-tremeGENE siRNA transfection reagent according to the manufacturer’s instructions in four GBM cell lines, namely, the SF188, T98G, GBM6, and GBM10. Following transfection for 24 h in the presence of DMEM with 10% FBS, the cells were trypsinized and processed for western blotting. A set of four shRNAs against the *CTCF* transcript cloned in psiLv-U6 plasmids, also coding for puromycin resistance was obtained from Genecopoeia (Rockville, MD, USA). Transient transfections of plasmid DNAs (1 µg) for 40 h were performed using the Roche X-tremeGENE HP reagent in 4 GBM cell lines. Cell extracts prepared from these experiments were processed for western blotting. A full-length expression vector of *NRF2* cloned in pCDNA3.1 was a gift from Dr. Anil Jaiswal (Baylor College of Medicine, Houston, TX, USA). After transfection of SF-188 glioblastoma cells by standard procedures and gentamycin selection, a clone expressing the highest level of NRF2 was isolated and cultured.

### 4.4. Chromatin Immunoprecipitation (ChIP) Assay

The EpiQuik ChIP assay kit (Epigentek, Farmingdale, NY, USA) was used to determine the presence of a CTCF-binding site in the MGMT-E1 promoter. Briefly, the SF-188 GBM cells were exposed to 1% formaldehyde in the medium without serum for 10 min, followed by lysis, DNA shearing, and protein/DNA Immunoprecipitation using CTCF antibodies (Cell Signaling Technology, Danvers, MA, USA). The antibodies to RNA polymerase were used as a positive control and the normal mouse IgG as a negative control. The purified DNA obtained after reversal of protein-DNA crosslinks by heating was subjected to SYBR green real-time q-PCR. The following set of primers was used to identify the *CTCF* motif in *MGMT* promoter 1: forward primer: GCCTTACCTCTAGGTGCCA, reverse primer: TGGCGGAGGGAAAGCTG. For determining the GAPDH promoter in RNA-polymerase enriched DNA, the primers were, forward GATTACGGGATGGGTCTGAA and reverse GCTGCACCTCTGGTAACTCC. The fold enrichment method was used to analyze the ChIP-qPCR data. The 2^−∆∆Ct^ is calculated from the outcome of (Ct IP)—(Ct mock). The normalization was done using the IgG Ct value.

### 4.5. Database Search

Genomics, promoter and protein databases, GenBank, Ensembl, EMBL-EBI, PrEESSTo/FANTOM5, UCSC genomic browser, TRED, EPD, and UniProt and Enzyme Portal were used to search for the *MGMT* genomic setting, regulatory elements, and the genomic criteria of the *MGMT-EBF3* genomic region, and MGMT protein sequence. The UTRdb research tool was used to identify the 5’-UTR and 3’-UTR regions of *MGMT* mRNA. The precise genomic map locations of the identified sequences were verified and updated to the hg38 version of the human genome sequence by using the BLAST tool/UCSC Genome Browser database. EMBOSS Needle tool was used for Pairwise Sequence Alignment of nucleotide sequences. The tools JASPAR and CTCFBSDB 2.0 [89,90] were used to identify the CTCF binding sites in the MGMT promoter and the rest of the gene.

### 4.6. Regulatory Sequences and MGMT Alternative Promoters

The transcription factor binding sites (TFBSs) were selected according to their unique functional properties that characterize the functional activity of the promoter, e.g., TATA, INF, DTIE, *CTCF, p53, MYC/MAX/MAD* genetic switch, and GAGA factor (GAF). Transcriptional Regulatory Element Database (TRED), JASPAR database, CTCFBSDB 2.0 database, and the Bio-Web tools were used to search for the TFBSs. Identification and plotting of CpG Islands in the *MGMT* promoter were achieved by EMBOSS CPGplot tool (https://www.ebi.ac.uk/Tools/seqstats/emboss_cpgplot/, accessed on 15 February 2021). The following parameter sets used to identify CpG islands in nucleotide sequence of MGMT promoter: Observed/Expected ratio > 0.60, Percent C + Percent G > 55.00, Length > 200 bp [91,92]. In-silico search was conducted to search for *MGMT* alternative promoters in NCBI/Nucleotide, PrESSTo/FANTOM, Ensembl 84, and EPD databases and to identify transcription factors binding sites (TFBSs) involved in the initiation of transcription and their setting in respect to CTCF motif in *MGMT* exon one revised promoter. The search included: TATA-8 (TATAWA, TATAWAWR) and TATA-532 (HWHWWWWR, excluding: HTYTTTWR, CAYTTTWR, MAMAAAAR and CTYAAAAR), INR (YYANWYY), DTIE (GBBRDNHGG), CCAAT and its inverted sequence TAACC, BRE (SSRCGCC), and DPE (RGWCGTG) binding sites [93].

### 4.7. Sequencing-Based Data from ENCODE and Bioinformatic Analysis of Non-Coding RNA Transcripts

Also analyzed were various sequencing-generated data from the ENCODE Project [94]. These include the transcription factor CHiP Seq [95], DNaseI hypersensitivity [96], and RNA sequencing [97] data that were accessed through the UCSC Genomics Browser. The following ENCODE project tier 1 cell lines [98] were analyzed for *MGMT* gene transcripts and NRF1 occupancy at the MGMT promoter: GM12878 (Lymphoblastoid), HeLa S3 (cervical cancer), K562 (chronic myelogenous leukemia), and H1 human embryonic stem cells (H1-hESC). Of these, the MGMT-proficiency of HeLa S3 cells [99] and its deficiency in K562 cells [100] has been reported in the literature.

Identification of un-reported mature sequences (miR) of ncRNA transcripts was performed by using the miRBase BLASTN search tool sequence option [101]. The miRNA seed sequences binding sites to target sequences (MRE) of *MGMT* and *EBF3* transcripts, were analyzed and identified by RNA22 v2 microRNA target detection tool [102]. The lower E-value or *p*-value represents, the greater chance that the loci contain a valid MRE.

### 4.8. Statistical Analysis

The statistical analyses of the obtained results were achieved using Excel software. T-test was used for the analysis of two Independent samples following ANOVA. One-way ANOVA was used to analyze the differences among group means of more than two samples. All statistical computations were calculated using GraphPad Prism 7. A significant difference was assessed at *p* < 0.05.

## Figures and Tables

**Figure 1 ijms-22-02492-f001:**
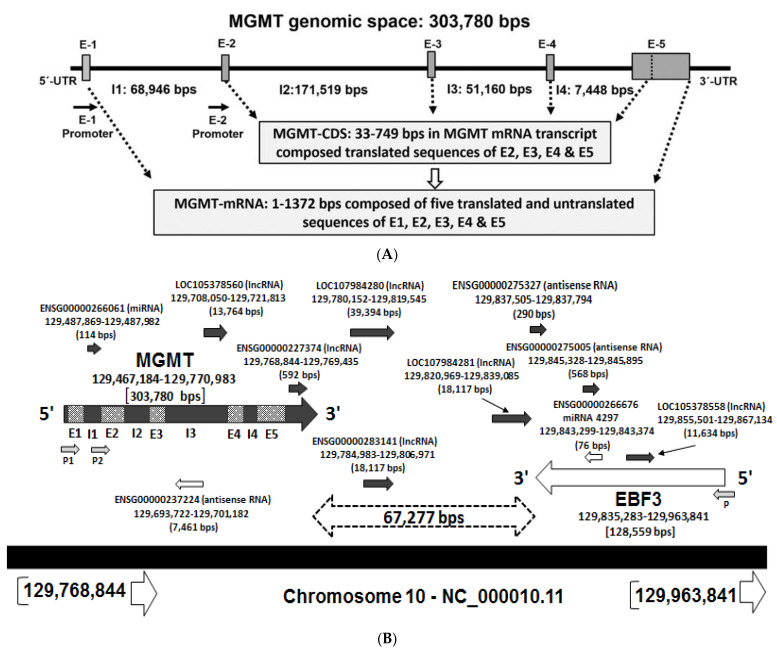
(**A**)—*MGMT* genomic space showing the two untranslated regions 5’-UTR and 3’-UTR, the five exons (E1–E5), four introns (I1–I4), two promoters (*MGMT-E1* and a putative *MGMT-E2* promoter identified in this report), and the coding sequence of MGMT mRNA transcript NM_002412. (**B**)—Schematic presentation of the MGMT-EBF3 genomic region along with ncRNA loci (miRNA, lncRNA, and antisense RNA) hosted in this space. Black arrows show the loci in the 5’–3’ direction on the forward strand. The white arrows indicate the loci in the 5’–3’ direction on the reverse strand. The bidirectional dotted arrow shows the space between *MGMT* and the *EBF3* genes. Letters “E” and “I” refer to exons and introns in the *MGMT* genomic space denoted by small light and dark gray boxes. P1 refers to the well-studied *MGMT* promoter proximal to exon-1. Our bioinformatic analysis showed the presence of a promoter-like sequence (indicated as P2) in the intron 1-exon 2 regions. LOC and ENSG refer to the loci IDs in the GenBank-Gene and Ensembl databases.

**Figure 2 ijms-22-02492-f002:**
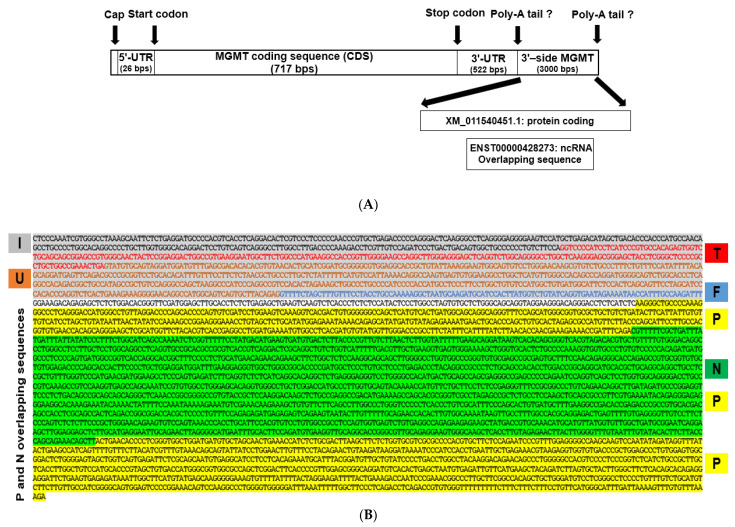
(**A**)—Schematic representation of the *MGMT* mRNA transcript, *MGMT* 5’-UTR, coding sequence, 3’-UTR region, and the added 3 kb sequence in the updated version of the GenBank-Gene that hosts the overlapping sequences of a discontinued protein-coding locus 105378559 transcript XM_011540451.1, and an ncRNA transcript, ENSG00000227374. (**B**)—The sequence of 3’-side of the *MGMT* locus showing the 3 kb of the two overlapping genes, uncharacterized locus 105378559 protein-coding (P) and RP11-109A6.3/ENSG00000227374 ncRNA gene (N). Letters I, T, U, and F respectively refer to the end of *MGMT* intron-4, the translated region of exon-5, the untranslated region of exon-5, and *MGMT* genomic feature respectively. The short non-highlighted sequence relates to the genomic region between the *MGMT* locus and the sequence of overlapping genes. (**C**)—Distribution of motifs in the established *MGMT* promoter X61657.1 mapped to chr10: 129466183 -129467339 [23]. The promoter was screened manually to locate the color-coded consensus recognition motifs. This is the first report describing ARE and ERE in the promoter. Note that the end of the AP1 recognition sequence overlaps with the ARE. The underlined sequence represents the untranslated *MGMT* exon 1.

**Figure 3 ijms-22-02492-f003:**
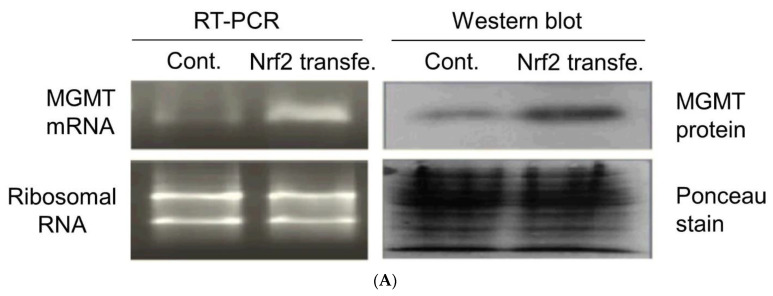
(**A**)—Augmented *MGMT* transcription in *NRF2* transfected SF-188 glioblastoma cells. Left panel—Total RNA isolated from the parent and transfected cells was subjected to RT-PCR using *MGMT*-specific primers. One µg RNA used in PCR products was electrophoresed on agarose gels at pH 8.0 and stained with ethidium bromide to show equivalent loading. Right panel–Western blot showing MGMT protein levels. The membrane stained before western blotting with Ponceau-S shows that protein loading was equal. (**B**)—The relationship between the transcription factor NRF1, DNaseI hypersensitivity (a measure of chromatin accessibility), and *MGMT* expression as depicted in a region of *MGMT* locus encompassing portions of the promoter, the entire exon 1, and the 5′ end of intron 1 are represented. As shown, the MGMT-proficient cell lines GM12878, H1-hESC, and HeLa-S3 were characterized by varying intensity of NRF1 bound at the *MGMT* promoter region. These cell lines also exhibited DNaseI hypersensitivity at this particular region of *MGMT* promoter besides direct evidence for *MGMT* expression (exon 1 read counts are shown). In contrast, the cell line MGMT-deficient K562 cells were devoid of bound NRF1 at its DNaseI insensitive promoter. No *MGMT* transcription was evident in K562 cells. The tracks were generated from UCSC Genome Browse using three types of ENCODE sequencing data: NRF1 ChIPSeq (University of Washington), paired-end RNA Seq (California Institute of Technology), and sequencing-based Digital DNaseI hypersensitivity assay (University of Washington).

**Figure 4 ijms-22-02492-f004:**
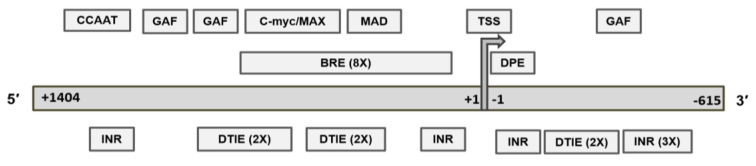
Upper panel—Locations of transcription factor binding motifs of genetic switch *c-myc/MAX/MAD* and transcription pausing factor (GAF) in the *MGMT-P1* revised promoter. BRE, B Recognition Element, DTIE, initiator of TATA-box minus transcription, INR, Initiator Element. Lower panel—Presence of p53 response elements (PREs) in the *MGMT* intron-1. The base pairs of PREs that perfectly conform to the p53 consensus motif are shown in underlined italic capital letters. The base-pair variations of the p53 consensus motif are shown in small letters. Key: Exon (E)), Intron (I), Open chromatin (O), CTCF (C), Enhancer (E), Promoter flank (F), Promoter (P). RRRCWWGYYY (N) RRRCWWGYYY, where R=A or G; W=A or T; Y=C or T; N (0–13) = A, T, C or G.

**Figure 5 ijms-22-02492-f005:**
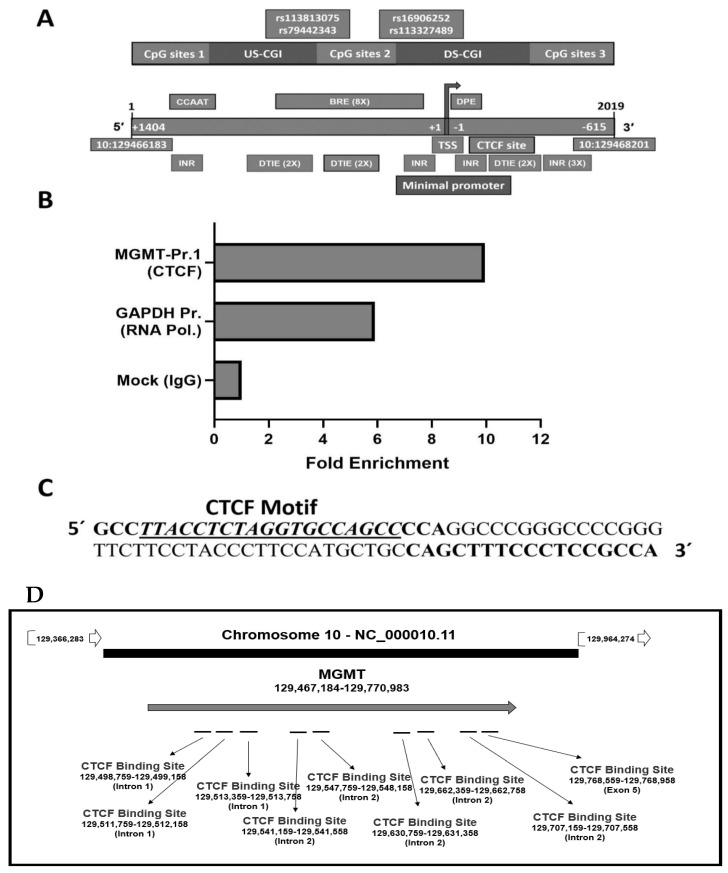
Identification of CTCF binding motifs in *MGMT* promoter, the body of the gene, and validation of its presence by Chip assay in the human *MGMT* revised exon-1 promoter. (**A**)—Mapping of the two CGIs (upstream and downstream of TSS) and in-silico identification of binding sites for CTCF and other transcription factors associated with initiation therein. INR—Initiator element which facilitates binding of transcription factor II D; DTIE and BRE recruit the TFIID and TFIIB respectively. DPE-Downstream Promoter Element. (**B**)—Identification of the CTCF binding sequence in the promoter by ChIP assay. CTCF and RNA Polymerase- bound genomic DNA fragments were isolated from the SF-188 glioblastoma cells using the respective antibodies. CTCF-bound DNA was amplified for the MGMT Promoter-1 and RNA-pol DNA for the GAPDH promoter (used as a positive control). Normal IgG alone served as a negative control. The PCR fragments were electrophoresed, photographed, and quantitated by densitometry. (**C**)—The sequence of 19 nucleotide CTCF binding site present in the *MGMT* promoter is shown. (**D**)—The distribution of nine CTCF binding sites present in *MGMT* genomic space as revealed by the Ensembl database search.

**Figure 6 ijms-22-02492-f006:**
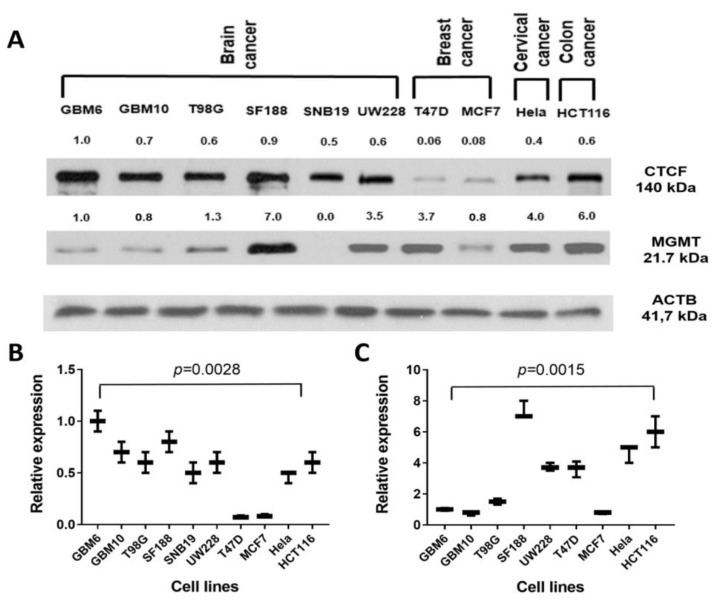
(**A**)—Western blots of CTCF and MGMT protein levels in human glioma and other cancer cell lines. (**B**)—ANOVA analysis of CTCF protein relative expression values in the ten cell lines, (**C**)—ANOVA analysis of MGMT protein relative expression values of nine cell lines, SNB19 excluded as it showed no MGMT expression.

**Figure 7 ijms-22-02492-f007:**
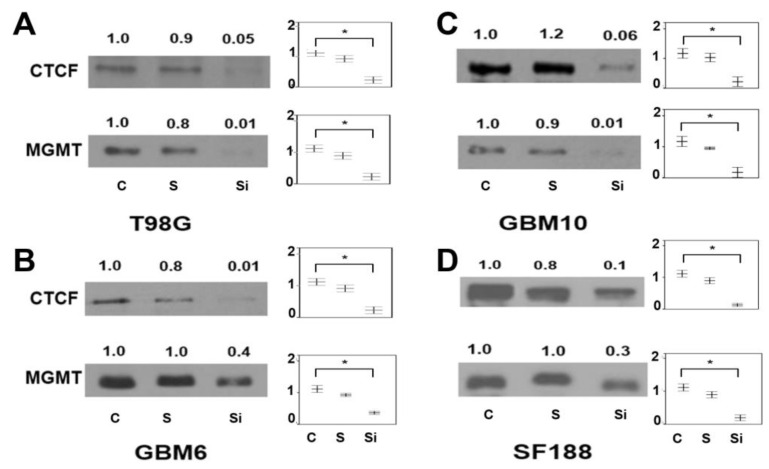
Significant downregulation of MGMT expression after silencing of the CTCF gene using siRNA targeted *CTCF* transcript variant 1 (NM_006565.3) in 4 glioblastoma cell lines shown in panels (**A**–**D**). Significant differences as determined by ANOVA are shown with an asterisk (*) from three independent experiments (*p* < 0.01).

**Figure 8 ijms-22-02492-f008:**
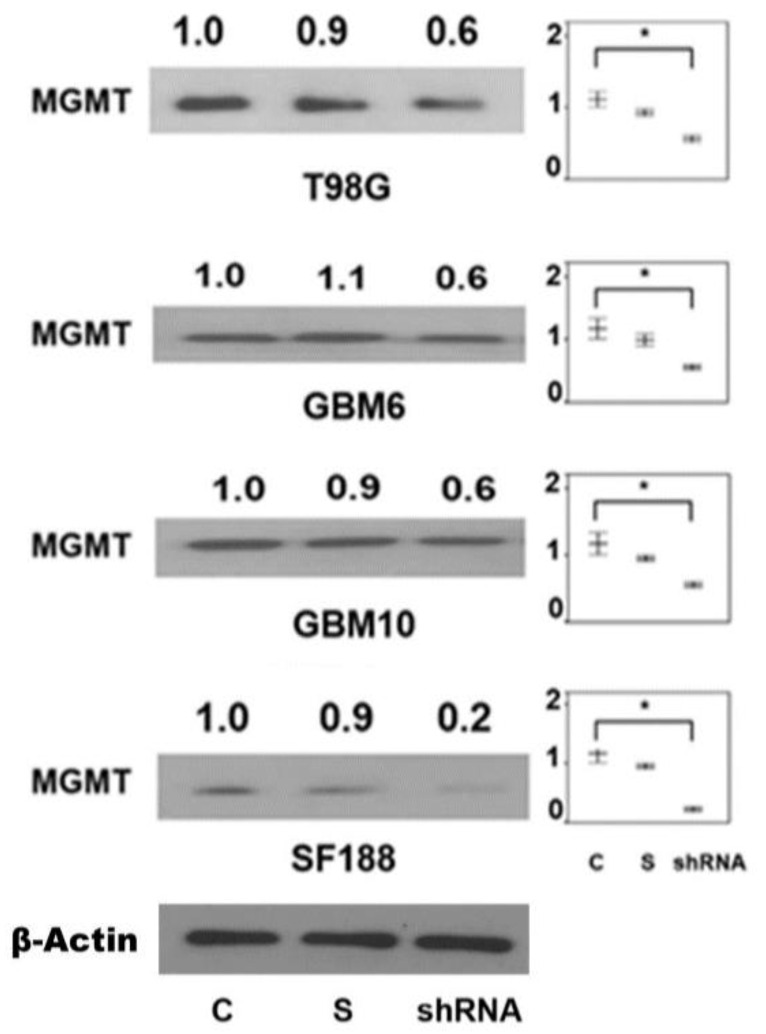
Marked downregulation of MGMT expression after silencing of the CTCF gene using an shRNA targeting the CTCF transcript variant 1 NM_006565.3 in 4 glioblastoma cell lines. Cells were transiently transfected with shRNA plasmids bearing CTCF-specific or a scrambled sequence for 24 h followed by western blotting. The mean expression of MGMT in 3 separate experiments was determined and shown in the side panels. Significant differences as determined by ANOVA are shown with an asterisk (*) from three independent experiments (*p* < 0.01).

**Table 1 ijms-22-02492-t001:** Five alternative overlapping promoters were identified at the 5’-side of *MGMT* exon-1. The revised version of the 5’-side MGMT exon-1 promoter (*MGMT-P1*) included the five sequences of overlapping alternative promoters. One predicted promoter was located at the 5’-side of *MGMT* exon-2 (*MGMT-P2*).

Source	Promoter ID	Map Locations at Chromosome 10/+ Strand	Span (bp)
PrESSTo/FANTOM	P1@MGMT	129,466,944–129,467,344	401
	P2@MGMT	129,466,905–129,467,305	401
Eukaryotic Promoter Database (EPD)	MGMT_1	129,466,745–129,467,344	600
NCBI/Nucleotide	X61657.1	129,466,183–129,467,339	1157
Ensembl 84	ENSR00001428452	129,466,558–129,468,201	1644
Revised MGMT exon-1 promoter (this study)	MGMT-P1	129,466,183–129,468,201	2019
Transcriptional Regulatory Element Database (TRED)	TRED-5071 (MGMT-P2)	129,535,540–129,536,539	1000

**Table 2 ijms-22-02492-t002:** Distribution of CpG islands (CGI) and GC content in revised *MGMT* exon-1 promoter (P1-CGI). US and DS refer to upstream and downstream sequences respectively. The CTCF binding sequence and TSS are located in DS-CGI.

CGI Annotation	Nucleotide Position 5’–3’	CGI Length (bp)	Map Location	%GC	Obs/Exp CpG
P1-CGI	1–2019	2019	chr10:129,466,183–129,468,201	63	0.63
US-CGI	241–720	480	chr10: 129,466,423–129,466,902	65.2	1.1
DS-CGI	901–1440	540	chr10:129,467,083–129,467,622	73.1	0.79

**Table 3 ijms-22-02492-t003:** Prediction of the human miRNA mature sequences (miR) in the ncRNA loci mapped at the *MGMT*-*EBF* region. Prediction is based on a search for similarity with miRBase mature miRNA sequences. The RNA22 v2 microRNA target detection tool was used to locate the predicted microRNA recognition elements (MRE) in the transcripts of *MGMT* (NM_002412.4) or *EBF3* (NM_001005463.2). The predicted miR sequences bind and target the MRE of *MGMT* and *EBF3* mRNAs. Abbreviations: miRNA: MicroRNAs, miR: mature miRNA product, hsa: human miRNA.

ncRNA	ncRNA Locus ID	ncRNA Transcript ID	The Similarity of Predicted miR with miRBase Mature miRNA Sequences	MRE Locations in MGMT and EBF3 mRNAs	No. of Identified MRE in *MGMT* and *EBF3* mRNAs	Leftmost Position (5’→3’) of Predicted *MGMT* and *EBF3* mRNAs Target Sites.
**miRNA**	ENSG00000266061	ENST00000585165.1	hsa-miR-574-5p	MGMT-3’-UTR	1	908
				EBF3-E8, E13, 3’-UTR	5	759, 1309, 3500, 3821, 4101
	MIR4297	ENST00000579857.1	has-miR-4297	MGMT-E4	1	406
				EBF3	0	-
**Antisense RNA**	ENSG00000275005	ENST00000614150.1	hsa-miR-4645-3p	MGMT	0	-
				EBF3-E2, E4	2	353, 423
	ENSG00000275327	ENST00000617939.1	hsa-miR-942-5p	MGMT-3’-UTR	1	1093
				EBF3-E5 & 3’-UTR	2	499, 3989
**IncRNA**	ENSG00000227374	ENST00000428273.1	hsa-miR-339-5p	MGMT	0	-
				EBF3-E1, E5, 3’-UTR	3	186, 505, 3290
	ENSG00000283141	ENST00000635764.1	hsa-miR-6862-5p	MGMT-3’-UTR	1	946
				EBF3	0	-

**Table 4 ijms-22-02492-t004:** Identification of miR sequences in noncoding RNA loci that bind target MREs of *MGMT* (NM_002412.4) mRNA transcript. Prediction of the mature miRNA sequence of a miRNA/ncRNA transcript by miRBase search tool and identification of miR binding sites of MGMT (NM_002412.4) microRNA recognition elements (MRE) and their corresponding heteroduplexes by RNA22 v2 microRNA target detection tool. Analysis conducted for prediction of mature sequences (miR) of miRNA loci (predicted ENSG00000266061 and miRNA4297), antisense RNA (ENSG00000275005 and ENSG00000275327), lncRNA (ENSG00000227374 and ENSG00000283141). MIR217 and MIR370 which are known to downregulate MGMT expression [79,85] were included for comparison. The upper sequence in the heteroduplex represents MRE for MGMT mRNA. The lower E-values and p-values mean the greater chance that the loci contains a valid MRE.

ncRNA Locus ID	Predicted miR by miRBase Search Tool				Predicted MRE by RNA22 v2 Tool	
	Sequence	hsa-miR	Score	E-value	Heteroduplex	*p* Value
ENSG00000266061	UGUGUGUGUGUGUGUGUGUGUGU	hsa-miR-574-5p	100	0.009	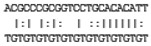	0.36
miRNA4297	UGCCUUCCUGUCUGUG	hsa-miR-4297	88	0.22	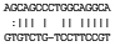	0.14
ENSG00000275005	UAGUUCUUGCCUGG	hsa-miR-4645-3p	70	4.2	-	-
ENSG00000275327	ACAUGGCCAAAACAGAG	hsa-miR-942-5p	67	3.6	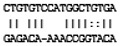	0.072
ENSG00000227374	AGGUUCCCUCUGGCCGC	hsa-miR-4726-3p	67	6.4	-	-
ENSG00000283141	GCAUGCUGGGAGAGACU	hsa-miR-6862-5p	67	2.4	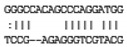	0.225
MIR127	UCGGAUCCGUCUGAGCUUGGCU	hsa-miR-127-3p	110	6 × 10^−4^	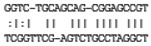	0.107
MIR370	GCCUGCUGGGGUGGAACCUGGU	hsa-miR-370-3p	110	7 × 10^−4^	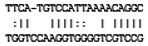	0.225

## Data Availability

Details of data from all databases and bioinformatic information presented in this study will be shared. Further, all cell lines and materials included here will be available for anyone interested.

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
