# Peer review of "Genomic Space of MGMT in Human Glioma Revisited: Novel Motifs, Regulatory RNAs, NRF1, 2, and CTCF Involvement in Gene Expression"

_ijms, 2021, doi:10.3390/ijms22052492_

Round 1
Reviewer 1 Report
The manuscript entitled „ Genomic Space of MGMT in Human Glioma Revisited: Novel Motifs, Regulatory RNAs, NRF1, 2 and CTCF Involvement in Gene Expression” provides the detailed and interesting review of regulatory elements of MGMT gene as well as brings new insights into the gene regulatory machinery.
General Comment:
The work was done with an attention and many regulatory aspects were shown. I have some minor comments which could be helpful in making text better.
MAJOR ISSUES
Please discuss your results of methylation level of MGMT specific regions in the respect to G-CIMP specific phenotype.
MINOR ISSUES
I got the impression that there are some double spaces – please doble check.
Please write gene symbols in italics – starting from the title.
Abstract
- c) two well-defined p53 response elements in MGMT intron-1 were identified. Please add a comma before “were identified”.
Introduction
- It is a large gene containing 5 exons, the first of which is non-coding [22] – please specify “large” in which sense? 5 exons is not much. Do you mean that it is a long gene? Or has long introns? Please be more specific here.
- In the sentence “MGMT promoter contains a 777-bp CpG island” please add “long” before “CpG island”.
- I suggest to change word “initiation” in the sentence “upstream of the transcription initiation site, including” to “start” – as the TSS (transcription start site) is a well-known term.
- “regions [15, 22] However” – insert a period – “regions [15, 22]. However”
- “transcription starts is separated” – I would suggest to change to “start”
2.1. Recapitulation of MGMT Genomic Space and the mRNA – the first paragraph of this section please shorten it and do not repeat information which are in the figure to which you refer.
- “1 and a large part of exon 5 are non-coding” add “the” – “1 and a large part of the exon 5 are non-coding”
- In the text “regulatory sequences in the MGMT-EFB3 genomic space.”, you create a specific term “MGMT-EFB3 genomic space” – please add it to the arrow in the figure 1B which shows this space. At the end of this sentence give the reference to the figure 1B.
-The last sentence of the 2.1. subchapter is unnecessary – the figure is well self-explanatory.
Subchapter 2.2.
- Figure 2A – why there are two question marks? They do not end the questions. If something is suggested name it “putative”. Please correct that.
- In this subchapter when referring to specific exons – the specific text formatting was added: capital letter, dash: Exon-1, Exon-5. I am not against such format but please introduce one format in the manuscript when referring to specific exons and later also introns. In the earlier chapters the formatting was different. I suggest not to name in the middle of sentence “exon” with capital letter.
- recent version of MGMT mRNA, Accession: NM_002412.5, to be the sequence of – change to “accession”
- an mRNA composed of two exons and a predicted – change to “of the two exons”
Subchapter 2.3.
- was recognized [23 ], - remove space
- promoter at -617 to -595 and – do not leave single dash at the end of the line. Make a stiff space between “–“ and the number
- at the end of figure 2 caption: “MGMT exon 1. . “ – remove one dot, make one format when referring to specific exons as mentioned earlier.
- “ERE regulatory site(Fig. 3B).” change to – site (Figure 3B)
- Figure 3 caption: start the caption on the left-hand side of the page.
- “tool showed 35 transcription starting sites” – change to “transcription start sites”
- “CTCF motif, to de discussed later in this report.” – “de” shall be changed to “be” I suppose
-There is no “Supplementary File: Figure S3A” and “Supplementary File: Figure S3B” – there is only Figure S3.
- The plot of the CGIs of MGMT exon-1 promoter GenBank-nucleotide “X61657.1” composed of 1157 bps hosted CGI of 621 bps (481-1101) – the plot showing that is S4 not S3 figure as referred in the text.
Subchapter 2.7
- a variable expression of these proteins. however, there – change to “However”
Conclusions:
- “consensus motif we report in intron 1 provides a mechanistic” – please refer to introns in the same manner as to exons – when you decide
4.7. Sequencing-based Data from ENCODE and Bioinformatic Analysis of Non-coding RNA Transcripts
Please add information about specific cell lines accordingly to specific data sets.
4.8. Statistical Analysis
Please add information whether you tested the normality distribution / homogeneity of variance of the data before using T-test and whether you verified if the data might be tested with ANOVA (normal distribution / homogeneity of variance).
In the section “Supplementary information:” in the main text – you write about three supplementary figures while in the supplementary file there are four figures.
Author Response
We thank the Reviewer for a detailed careful reading of our manuscript, the positive comments, and constructive suggestions. The errors pointed out throughout the text helped improve the manuscript. The revised manuscript Specific responses to the questions/comments are listed:
MAJOR ISSUES:
Please discuss your results of the methylation level of MGMT specific regions in the respect to G-CIMP specific phenotype.
Response: Since MGMT promoter methylation is a focal point in human gliomas, we fully agree this is a very valid point. In response, we have expanded the Introduction to describe the prevalence of CIMP in gliomas (G-CIMP), the IDH mutations, the oncometabolite D- 2HG, the mechanism by which D2HG manifests the G-CIMP (pages 2-3), and discussion on a) coexistence of MGMT promoter methylations with IDH mutations and b) how our findings of CTCF binding in the MGMT genome may be disrupted because of CpG methylation (end of page 8). Appropriate references (26-30,69, 71, 74, 75) have been included.
MINOR ISSUES:
I got the impression that there are some double spaces – please double check
Thank you. This has been checked.
Please write gene symbols in italics – starting from the title.
Throughout the manuscript, we have now italicized the gene symbols when they refer to the gene and non-italicized when they refer to the protein in the context of statements.
Abstract: - c) two well-defined p53 response elements in MGMT intron-1 were identified. Please add a comma before “were identified”.
Comma has been added.
Introduction:
- It is a large gene containing 5 exons, the first of which is non-coding [22] – please specify “large” in which sense? 5 exons is not much. Do you mean that it is a long gene? Or has long introns? Please be more specific here.
The statements have been better written to reflect the large gene size, longer introns, and shorter exons (page 2, para 2).
- In the sentence “MGMT promoter contains a 777-bp CpG island” please add “long” before “CpG island”.
“long” added as suggested (page 2).
- I suggest changing the word “initiation” in the sentence “upstream of the transcription initiation site, including” to “start” – as the TSS (transcription start site) is a well-known term.
This has been changed to TSS (page 2).
- “regions [15, 22] However” – insert a period – “regions [15, 22]. However” –
A period has been inserted.
- “transcription starts is separated” – I would suggest changing to “start”
This has been changed to ‘start’ (page 3, para 2).
2.1. Recapitulation of MGMT Genomic Space and the mRNA – the first paragraph of this section please shorten it and do not repeat information which is in the figure to which you refer.
This section has been shortened and repetition removed (page 3, para 3).
- “1 and a large part of exon 5 are non-coding” add “the” – “1 and a large part of the exon 5 are non-coding”
“the” added as suggested (page 3, para 3).
- In the text “regulatory sequences in the MGMT-EFB3 genomic space.”, you create a specific term “MGMT-EFB3 genomic space” – please add it to the arrow in the figure 1B which shows this space. At the end of this sentence give the reference to the figure 1B.
Figure 1B has been added at the end of the sentence. The legend to Figure 1B describes the “MGMT-EBF3 genomic space”
-The last sentence of the 2.1. subchapter is unnecessary – the figure is well self-explanatory.
Agree, the last sentence has been deleted (page 3, penultimate paragraph).
Subchapter 2.2.-
- Figure 2A – why there are two question marks? They do not end the questions. If something is suggested name it “putative”. Please correct that.
This appears to have been corrected by the journal editing staff, thank you.
- In this subchapter when referring to specific exons – the specific text formatting was added: capital letter, dash: Exon-1, Exon-5. I am not against such format but please introduce one format in the manuscript when referring to specific exons and later also introns. In the earlier chapters, the formatting was different. I suggest not to name in the middle of the sentence “exon” with a capital letter.
Yes, agreed. The word exon starts with a capital letter only at the beginning of a sentence throughout the manuscript.
- recent version of MGMT mRNA, Accession: NM_002412.5, to be the sequence of – change to “accession”
Changed as suggested.
- an mRNA composed of two exons and a predicted – change to “of the two exons”
Changed to “of two exons” (page 4, para 1).
Subchapter 2.3.
- was recognized [23 ], - remove space
Space has been removed.
- promoter at -617 to -595 and – do not leave a single dash at the end of the line. Make a stiff space between “–“ and the number
Modified as suggested (page 4, para 2).
- at the end of figure 2 caption: “MGMT exon 1. . “ – remove one dot, make one format when referring to specific exons as mentioned earlier.
Changed as suggested.
- “ERE regulatory site(Fig. 3B).” change to – site (Figure 3B)
Yes, has been changed to Figure 3B from Fig. 3B (page 5, para 2).
- Figure 3 caption: start the caption on the left-hand side of the page.
Yes, the caption has been shifted to the next line.
- “tool showed 35 transcription starting sites” – change to “transcription start sites”
Agree, this has been changed to “transcription start sites” (last sentence, page 5).
- “CTCF motif, to de discussed later in this report.” – “de” shall be changed to “be” I suppose
Thank you. The typo has been corrected.
-There is no “Supplementary File: Figure S3A” and “Supplementary File: Figure S3B” – there is only Figure S3.
The reviewer is correct. Sorry for the error in Figure numbering. However, now, there are Figures S3A and S3B, they are correctly labeled.
- The plot of the CGIs of MGMT exon-1 promoter GenBank-nucleotide “X61657.1” composed of 1157 bps hosted CGI of 621 bps (481-1101) – the plot showing that is S4 not S3 figure as referred in the text.
Please see the response above. Figure numbers have been corrected now.
Subchapter 2.7
- a variable expression of these proteins. however, there – change to “However”
Thank you. The sentence now starts as ‘However …….’ (page 8).
Conclusion:
- “consensus motif we report in intron 1 provides a mechanistic” – please refer to introns in the same manner as to exons – when you decide
Thank you. This has been corrected.
4.7. Sequencing-based Data from ENCODE and Bioinformatic Analysis of Non-coding RNA Transcripts
Please add information about specific cell lines accordingly to specific data sets.
Indeed, information on these cell lines (ENCODE project tier 1 cell lines) and their MGMT proficiency have been added along with pertinent references (page 13 under section 4.7.).
4.8. Statistical Analysis
Please add information whether you tested the normality distribution / homogeneity of variance of the data before using T-test and whether you verified if the data might be tested with ANOVA (normal distribution / homogeneity of variance).
The datasets were verified for ANOVA eligibility and the ANOVA-tested data was subjected to T-tests. Statement to this effect has been made (page 13).
In the section “Supplementary information:” in the main text – you write about three supplementary figures while in the supplementary file there are four figures.
Thank you. Supplementary information has now been corrected.
Reviewer 2 Report
Although the authors depict extensive bioinformatic analysis on MGMT promotor sequence in the present study, the results extremally lack of sufficient data validation. Almost all conclusions are brought on these hypothesies, and are in need for further exploration. Therefore, the scientific significance of present data remain unclear. In addition, as the authors demonstrate a correlation between expression of CTCF and MGMT (this is a novel result), overall this observation lack of scientific significance.
Author Response
Although the authors depict extensive bioinformatic analysis on MGMT promoter sequence in the present study, the results extremely lack of sufficient data validation. Almost all conclusions are brought on these hypothesies, and are in need for further exploration. Therefore, the scientific significance of present data remain unclear. In addition, as the authors demonstrate a correlation between expression of CTCF and MGMT (this is a novel result), overall this observation lack of scientific significance.
The authors agree on some points made by the reviewer. However, we want to make this clear -MGMT remains therapeutically the most important gene in glioma and other human cancers. The knowledge on the MGMT gene is so far limited to its silencing by promoter methylation. Apart from the G-CIMP, there has been practically no other data on this gene, its transcription, and how its promoter methylation is mechanistically manifested in the last two to three decades. This report aimed to bring new information on motifs, promoter elements, 3’UTR, and regulatory RNAs at the locus together. Providing experimental evidence for all observations made is way beyond the scope of this paper. We are happy to see that the reviewer feels the CTCF-MGMT connection is great. Our report has indeed provided evidence that CTCF is an important player in MGMT expression and interfering with CTCF expression will curtail MGMT expression. In summary, we believe the data presented here has great promise to open up new avenues of scientific exploration on MGMT genomics and hopefully to MGMT-directed therapeutics as well.
Round 2
Reviewer 2 Report
As my previous statements on the present manuscript were a bit harsh, the authors confidently convinced me, that the concerns are a bit out of present manuscript scope. I agree that all results exposed in the present study could light up a further valuable research. Good luck!